# A mesoscopic simulator to uncover heterogeneity and evolutionary dynamics in tumors

Juan Jiménez-Sánchez[1]☯*, Álvaro Martínez-Rubio [1,2,3]☯, Anton Popov[1], Julián Pérez-Beteta [1], Youness Azimzade [4], David Molina-García [1], Juan Belmonte-Beitia [1], Gabriel F. Calvo [1], Víctor M. Pérez-García [1]

**1** Department of Mathematics, Mathematical Oncology Laboratory (MOLAB), Universidad de Castilla-La Mancha, Ciudad Real, Spain, **2** Department of Mathematics, Universidad de Cádiz, Cádiz, Spain, **3** Biomedical Research and Innovation Institute of Cádiz (INiBICA), Cádiz, Spain, **4** Department of Physics, University of Tehran, Tehran, Iran

☯ These authors contributed equally to this work.
* Juan.JSanchez@uclm.es

**Data Availability Statement:** All relevant data are within the manuscript. The mesoscopic simulator code is available at https://github.com/JuanJS117/MesoscopicModel.

## Abstract

Increasingly complex in silico modeling approaches offer a way to simultaneously access cancerous processes at different spatio-temporal scales. High-level models, such as those based on partial differential equations, are computationally affordable and allow large tumor sizes and long temporal windows to be studied, but miss the discrete nature of many key underlying cellular processes. Individual-based approaches provide a much more detailed description of tumors, but have difficulties when trying to handle full-sized real cancers. Thus, there exists a trade-off between the integration of macroscopic and microscopic information, now widely available, and the ability to attain clinical tumor sizes. In this paper we put forward a stochastic mesoscopic simulation framework that incorporates key cellular processes during tumor progression while keeping computational costs to a minimum. Our framework captures a physical scale that allows both the incorporation of microscopic information, tracking the spatio-temporal emergence of tumor heterogeneity and the underlying evolutionary dynamics, and the reconstruction of clinically sized tumors from high-resolution medical imaging data, with the additional benefit of low computational cost. We illustrate the functionality of our modeling approach for the case of glioblastoma, a paradigm of tumor heterogeneity that remains extremely challenging in the clinical setting.

## Author summary

Computer simulation based on mathematical models provides a way to improve the understanding of complex processes in oncology. In this paper we develop a stochastic mesoscopic simulation approach that incorporates key cellular processes while keeping computational costs to a minimum. Our methodology captures the development of tumor heterogeneity and the underlying evolutionary dynamics. The physical scale considered allows microscopic information to be included, tracking the spatio-temporal evolution of

**Funding:** This research has been supported by grants awarded to VMPG by James S. Mc. Donnell Foundation, United States of America, 21st Century Science Initiative in Mathematical and Complex Systems Approaches for Brain Cancer (collaborative award 220020560) and Junta de Comunidades de Castilla-La Mancha, Spain (grant number SBPLY/17/180501/000154). VMPG and GFC thank the funding from Ministerio de Ciencia e Innovación, Spain (grant number PID2019-110895RB-I00). This research has also been supported by a grant awarded to GFC and JBB by the Junta de Comunidades de Castilla-La Mancha, Spain (grant number SBPLY/19/180501/000211). AMR received support from Asociación Pablo Ugarte (http://www.asociacionpablougarte.es). JJS received support from Universidad de Castilla-La Mancha (grant number 2020-PREDUCLM-15634). The funders had no role in study design, data collection and analysis, decision to publish, or preparation of the manuscript.

**Competing interests:** The authors have declared that no competing interests exist.

tumor heterogeneity and reconstructing clinically sized tumors from high-resolution medical imaging data, with a low computational cost. We illustrate the functionality of the modeling approach for the case of glioblastoma, an epitome of heterogeneity in tumors.

## Introduction

Discrete mathematical models in cancer track and update individual cells according to a set of biological rules as they interact with other cells and the microenvironment. There is a wide variety of models of this type that include both on-lattice (such as cellular automata) and off-lattice (agent-based) models. With the advent of single-cell resolution technology, next-generation sequencing techniques and the increasing availability of patient data, many mathematical modeling efforts in oncology have been directed towards the use of discrete and individual-based methodologies (see e.g. [1–4] for some reviews). These types of models are being used to address a broad variety of cancer-related problems and some of them are even available as open platforms for broad purpose simulation in cancer [5, 6].

However, discrete individual-based models also have some shortcomings. They typically incorporate many parameters that have to be obtained from a limited amount of biological information/data. Also, they are difficult to connect with imaging data, since medical imaging has a limited spatial resolution of about 1 mm$^3$. Although imaging techniques provide rough information on tumor cell density, metabolic activity, vascular status, and other relevant variables, they do include the details of cellular dynamics within each voxel. Thus, there is room for discrete cell-based modeling approaches beyond classical continuous ones based on partial differential equations but working at the spatial scales at which tumor evolution can be monitored.

From the computational point of view, individual-based models are computationally intensive and suited to describing microscopic scenarios, in vitro experiments, or even small sections of tissues or model animals. Addressing human tumors in clinical stages normally involves reducing the number of interactions, individuals or processes considered at the microscopic levels, or representing them in a simpler manner. While awaiting progress in computational power that allows for the inclusion of both detailed single cell information and macroscopic simulated tumors, the choice of scale seems to be the dominant factor [7]. As has recently been pointed out [8], few models are able to take into account three-dimensional space, a broad mutational spectrum, mixing populations and reaching clinical or realistic sizes in feasible computational time.

In this paper we put forward a three-dimensional, mesoscale, discrete, on-lattice, multi-compartmental, stochastic approach intended to simulate biological phenomena in clinically-sized tumors. The main element or agent is the cell subpopulation, whose definition is parallel to how species are normally defined in ecological models [9]. Space is discretized in compartments of adjustable size, which allows for comparison with medical imaging data. A compartment is occupied by a number of cell subpopulations with different features, each undergoing dynamics in its spatial location: growth, interaction with others and spatial spreading. This intermediate scale allows for the integration of detailed biological data and for computationally feasible simulations up to the macroscopic, whole-tumor scale.

The global evolution of the tumor is driven by the dynamics at each lattice position which, in turn, is governed by the behavior of the different cell subpopulations, which are subject to four fundamental biological processes: reproduction, death, mutation and migration. These four processes, stemming from the basic hallmarks of cancer [10], occur stochastically in a

synchronous manner, meaning that at each time step the population is updated according to the probabilities computed in the previous step. As a result, the system moves from a very simple initial state to a fully grown, realistically sized, heterogeneous tumor, reconstructing its entire natural history. Thus, the intent of this work is to present a computational framework to study evolutionary dynamics of tumors. We provide an explanation of the functionality and access to the codes. Owing to its low computational cost and simplicity, it allows for a comparison with clinical information, revealing its potential applications in hypothesis generation and testing.

As a bench test to assess our model's performance and versatility in tumor modeling, we also show a detailed application of the mesoscopic model to glioblastoma (GBM), one of the most aggressive tumors, which also epitomizes intratumor heterogeneity and enhanced phenotypic adaptation capacity [11], with only approximately one in four patients alive two years after diagnosis using current treatment modalities [12, 13]. By means of the TCGA genomic characterization of GBM [14], information about the mutational spectrum of this tumor [15] and the relative frequencies [16], coexistence [17] and exclusivity [18] is now available, sometimes including spatial information [19] and reconstructions of its evolutionary history [20, 21]. Imaging data is also increasingly available, providing valuable details related to geometry, shape, size, regularity, and it is also used in biomarker identification [22–24]. It is therefore an ideal scenario for testing and calibrating a model for tumor growth and diversification that includes molecular information and reaches clinical sizes.

## Materials and methods

### Computational model

**3D lattice.**   Space is discretized as a hexahedral mesh consisting of $L_x \times L_y \times L_z$ spatial units (voxels) or compartments, with $L_i$ being the number of compartments per spatial dimension. Both the volume and number of compartments in the grid are adjustable parameters. Since high-resolution imaging (e.g. 3D magnetic resonance imaging T1/T2/FLAIR sequences) voxel size is around 1 mm$^3$, we chose that voxel size for the specific examples. No-flux boundary conditions were set. However, we chose lattices large enough so that simulations will typically not reach the boundaries of the domain.

**Clonal subpopulations.**   Each voxel contains cells that undergo different cellular processes: division, migration, death and mutation. A cell belongs to a clonal subpopulation that is defined by a set of traits. Traits are represented by a vector $\vec{g}$ of length $G$, $\vec{g} = (g_1, g_2, ..., g_G)$, where $G$ is the number of traits or alterations and $g_i$ can take the values 0 or 1. This value can be interpreted as two possible expressions for a trait, or presence or absence of a mutation, as typically done when treating species in ecological models [9]. The clonal subpopulation then constitutes the basic unit or agent in the computational model, trading cell resolution for feasible simulation time and achieving clinical tumor sizes. The set of traits that represents a subpopulation determines the rate at which cells undergo biological processes, so that cells from a given subpopulation on a given voxel will behave in the same way, except for stochastic noise intrinsic to cell processes (biological instances of this are differential gene expression or variable mitochondrial content). Populations with more advantageous traits will be more likely to become fixed in the tumor, especially once they achieve a large cell number, whereas at early stages (low cell numbers) genetic drift will be more important. The processes are modeled in such a way that cells grow exponentially when there is enough empty space and slow their growth as the voxel becomes crowded. Migration is also influenced by voxel occupation as described below.

**System updating.** In celular automaton and lattice gas models, the way in which the spatial domain is updated can impact the outcome of the simulation. In [25], the authors discussed different manners of iterating through the grid. They recommended asynchronous updating, drawing the time of event from an exponential distribution and randomly choosing the lattice point to be updated. However, for processes with long timescales such as tumor development, it is reasonable to replace this with synchronous updating with uniform time step. In this manner, we compute the change in the population of each voxel according to the state at time $t$, and then add these changes to obtain the state at $t + 1$. Thus, if we let $N_{g,t}^{x,y,z}$ be the number of cells in subpopulation $g$ at voxel $(x, y, z)$ and time $t$, for $g = 1, \ldots, 2^G$ and $x = 1, \ldots, L_x$, $y = 1, \ldots, L_y$, $z = 1, \ldots, L_z$, the number of cells at the discrete time $t + 1$ can be computed using the balance equation

$$
\begin{aligned}
N_{g,t+1}^{x,y,z} &= N_{g,t}^{x,y,z} + N_{g_{\mathrm{rep}},t}^{x,y,z} - N_{g_{\mathrm{death}},t}^{x,y,z} - N_{g_{\mathrm{mut}},t}^{x,y,z} + N_{g'_{\mathrm{mut}},t}^{x,y,z} - N_{g_{\mathrm{mig}},t}^{x,y,z} \\
&\quad + \sum_{x',y',z' \in \mathcal{M}_{x,y,z}} N_{g_{\mathrm{mig}},t}^{x',y',z'}.
\end{aligned}
\tag{1}
$$

Eq (1) governs the updating of the cell number of each clonal population according to the four basic cellular processes described above. It includes the positive contributions of newborn cells $N_{g_{\mathrm{rep}},t}^{x,y,z}$, mutations from other clonal populations that come to the one evaluated $N_{g'_{\mathrm{mut}},t}^{x,y,z}$ and cells in the same subpopulation migrating from other voxels in the neighborhood $\mathcal{M}_{x,y,z}$ of current point, $N_{g_{\mathrm{mig}},t}^{x',y',z'}$. Cell numbers decrease by subtracting dead cells $N_{g_{\mathrm{death}},t}^{x,y,z}$, cells mutating to different clonal subpopulations $N_{g_{\mathrm{mut}},t}^{x,y,z}$ and cells migrating to surrounding voxels $N_{g_{\mathrm{mig}},t}^{x,y,z}$. Each process has an associated probability that depends on both local conditions and phenotype. A cell attempting to undergo any of these processes may be regarded as a dichotomous event with two possible outcomes: success (cell dividing, dying, migrating or mutating) or failure (cell remaining still). Hence, each cell can be considered as a Bernoulli experiment. The way in which the model is discretized allows us to assume that cells in the same voxel and subpopulation will have the same probability. We can therefore update the whole clonal subpopulation of a voxel by means of the binomial distribution, as it will give us the number of successes from several Bernoulli trials (cells undergoing basic processes) that share the same probability. The only exception is the mutation process: since its timescale is much longer than that of cell division or death, mutations (being Bernoulli processes) are assumed to occur in individual cells instead of sampling them from the binomial distribution. Following these considerations, we can write

$$
N_{g_{\mathrm{rep}},t}^{x,y,z} \sim \mathrm{B}(A_{g,t}^{x,y,z}, P_{\mathrm{rep}}),
\tag{2a}
$$

$$
N_{g_{\mathrm{mig}},t}^{x,y,z} \sim \mathrm{B}(A_{g,t}^{x,y,z}, P_{\mathrm{mig}}),
\tag{2b}
$$

$$
N_{g_{\mathrm{death}},t}^{x,y,z} \sim \mathrm{B}(A_{g,t}^{x,y,z}, P_{\mathrm{death}}),
\tag{2c}
$$

$$
N_{g_{\mathrm{mut}},t}^{x,y,z} \sim \mathrm{Bernoulli}(P_{\mathrm{mut}}).
\tag{2d}
$$

where $A_{g,t}^{x,y,z}$ represents the number of living cells in subpopulation $g$ at voxel $(x, y, z)$ and time $t$ and $P_{\mathrm{rep}}$, $P_{\mathrm{mig}}$, $P_{\mathrm{death}}$ and $P_{\mathrm{mut}}$ denote the single cell probabilities of division, migration, death and mutation, respectively. These probabilities are specified below. A summary of the updating algorithm can be found in Fig 1. More details about the coding of the algorithm can be found at the model's Github repository (see 'Software').

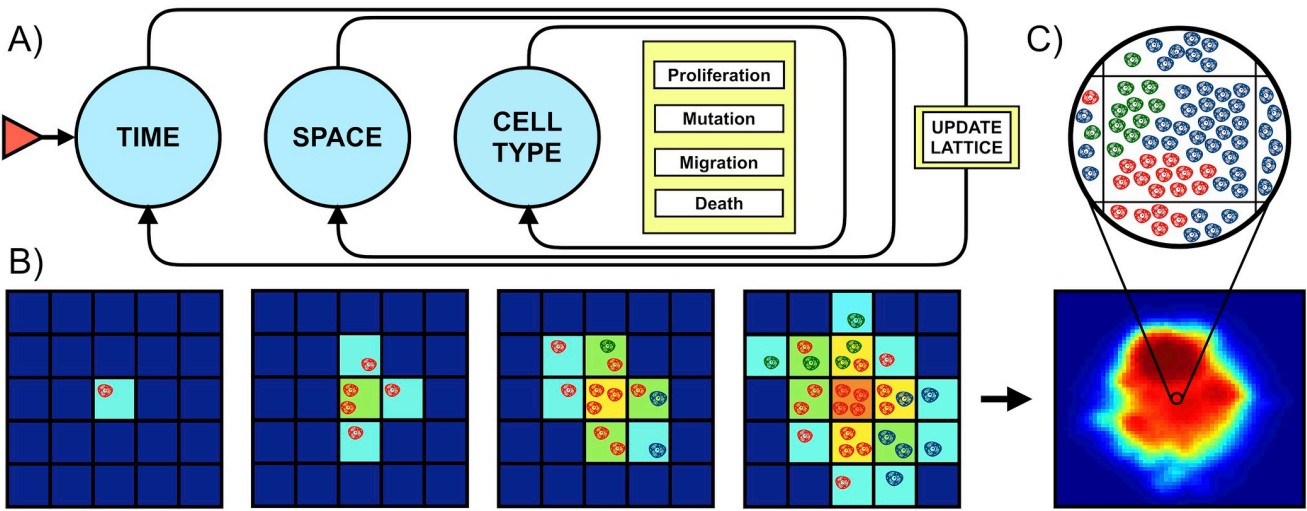

**Fig 1. Algorithm. A)** Basic algorithm (for implementation purposes). Initialization requires creating a 3D grid, specifying the final simulation time, defining subpopulation traits and setting the initial state. Temporal iterations are then carried out until the end time is reached. At each time step and each voxel, all clonal populations are updated. This updating involves calculating how many cells will proliferate, migrate, mutate or die. When all populations at all voxels have been evaluated, they are updated synchronously. **B)** Two-dimensional example of model behavior. Synchronous updating results in population increasing, diversifying and spreading at each time step, with probabilities computed according to the biological rules. Compartment color indicates occupancy. Cell color indicates cell type. **C)** Bottom image is a slice of an actual simulation, with colors indicating occupation. Each voxel contains a variable number of cells and subtypes as depicted above.

**Cell division.** Each clonal subpopulation has a different reproduction probability depending on its traits and its local (voxelwise) environment. Mathematically, this probability is expressed as:

$$P_{\text{rep}} = \frac{\Delta t}{\tau^{\text{rep}}}\left(1 - \frac{A + D}{N_{\text{max}}}\right),$$

$$(3)$$

where $\Delta t$ is the time step considered, $A$ and $D$ are the total active and necrotic population in each voxel respectively and $N_{\text{max}}$ is the local carrying capacity. The probability is modulated by the relationship between the time step and the characteristic time of the process; the probability of a process to occur increases with the time step. The reproduction probability decreases with occupation, simulating competition for space and resources. $\tau^{\text{rep}}$ is the part of the probability that depends on each subpopulation's traits. It has units of time and is computed as

$$\tau^{\text{rep}} = \tau_0^{\text{rep}}(1 - \vec{w}^{\text{rep}} \cdot \vec{g}),$$

$$(4)$$

where the first term, $\tau_0^{\text{rep}}$, is a basal reproduction time, assigned to the wild type, and the second term represents how this basal rate is modified by the different alterations that the cell can undergo. Also, $\vec{g}$ is the trait vector and $\vec{w}^{\text{rep}} = (w_1, w_2, ..., w_G)$, with $w_i$ representing the degree to which an alteration modifies the respective rate. It satisfies $|w_i| < 1$ and $\sum_i^G w_i < 1$.

**Cell migration.** The migration process occurs in two differentiated steps. Firstly, the number of cells leaving the current voxel is computed. They are then distributed into neighboring voxels, taking into account their relative distances. The probability of migration for a single cell is given by

$$P_{\text{mig}} = \rho^{\text{mig}} \frac{\Delta t}{\Delta x^2}\left(\frac{A + D}{N_{\text{max}}}\right).$$

$$(5)$$

Eq (5) has a similar structure to the reproduction process in Eq (3) albeit the trait-dependent term $\rho^{\mathrm{mig}}$ has the units of a diffusion coefficient (rather than being the inverse of a characteristic time), and differs from the impact of each alteration on the process, encompassed by the vector $\vec{w}^{\mathrm{mig}}$. We include here the spatial step $\Delta x$, by analogy with the discrete Laplacian operator in space, in order to keep the probability adimensional. We calculate $\rho^{\mathrm{mig}}$ in the following recursive form, similar to (4), as

$$\rho^{\mathrm{mig}} = \frac{\rho_0^{\mathrm{mig}}}{\left(1 - \vec{w}^{\mathrm{mig}} \cdot \vec{g}\right)} \ , \tag{6}$$

where $\rho_0^{\mathrm{mig}}$ is a basal diffusion constant corresponding to the wild type modulated by subsequent alterations. Notice that the product between $\vec{w}^{\mathrm{mig}}$ and $\vec{g}$ is moved to the denominator in order to keep the same structure as in the other basic processes. In contrast with the reproduction probability, the migration probability increases with the occupation of the voxel, since a cell is more likely to migrate if there is more competition for space and/or resources. Necrotic cells do not migrate, but they do occupy space, and thus this is taken into account in (5).

As for the destination of each cell migrating from a voxel $(x, y, z)$, we considered a Moore neighborhood $\mathcal{M}_{x,y,z}$ in three spatial dimensions. In this way, each migrating cell has 26 possible destinations. In order to determine which neighbouring voxel receives which number of cells, we give a probability for each neighbor. This probability will be proportional to $1, 1/\sqrt{2}$ or $1/\sqrt{3}$ depending on whether voxels share a face, edge or vertex with the central voxel respectively. The distribution of migrating cells into each destination voxel is then computed by sampling a multinomial distribution:

$$\vec{Y} \sim \mathrm{Mult}\left(X, \vec{P}_{\mathrm{Moore}}\right)$$

where $\vec{Y}$ is a vector of 26 components giving the number of cells that migrate to each voxel, $X$ is the number of migrating cells that has been previously computed according to Eq (2b) and $\vec{P}_{\mathrm{Moore}}$ is a vector of 26 components giving the probabilities of migration to each surrounding voxel. Performing the migration in this way reproduces a diffusive process, in which migration depends on cell density gradients and distances. To better illustrate migration, a scheme of the process can be seen in S1 Appendix.

**Cell mutation.**   Mutation is the mechanism that captures the diversification of the population, and is considered here in a broad sense as a change in one of the characteristics of the cell, be it genetic or phenotypical. The mathematical formula used to compute the mutation probability is

$$P_{\mathrm{mut}} = \frac{\Delta t}{\tau^{\mathrm{mut}}} \left(\frac{A}{N_{\mathrm{max}}}\right), \tag{7}$$

so that mutations depend only on $A$, the number of cells in a given subpopulation; $N_{\mathrm{max}}$, the carrying capacity, and $\tau^{\mathrm{mut}}$, the characteristic time that includes trait effects as explained above. This probability measures the chance of a mutational event occurring, not the specific alteration; this is later randomly selected from a subset of the vector $\vec{g}$, representing non-altered traits.

**Cell death.**   The form of the cell-death term is similar to the proliferation term, with the difference that the probability of death increases with occupation. The expression for the

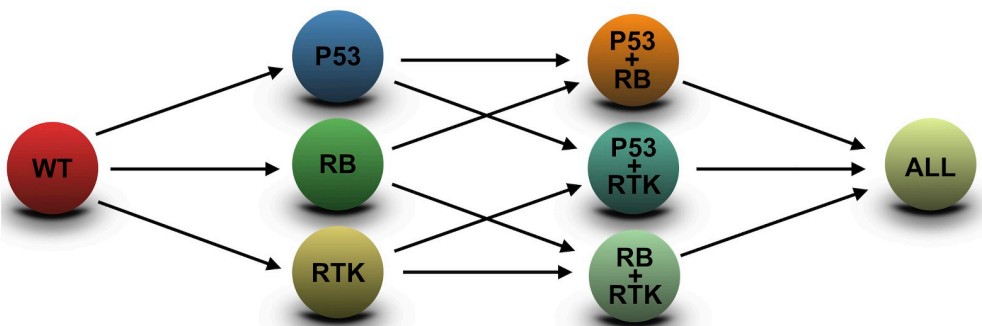

**Fig 2. Mutation tree used in this paper.** Relationships between the eight possible genotypes, according to the three alterations selected. Each clonal type can emerge from several ancestors by various alterations. Depending on the mutational history, tumors follow different paths on the mutation tree.

probability of death is then given by

$$P_{\text{death}} = \frac{\Delta t}{\tau^{\text{death}}} \left( \frac{A + D}{N_{\max}} \right),$$

(8)

where $\tau^{\text{death}}$ is a typical cell lifetime. The reason for this dependence is that aggressive tumor cells eventually induce a damage to the microenvironment that leads to their own death. In the specific tumor type considered in this paper, GBM, it is well known that this happens through the secretion of prothrombotic factors that promote microvessel failure and the formation of necrotic areas [26].

## Estimation of parameters

Since one of our goals is to apply the mesoscopic simulator to the case of GBM as a benchmark test, we used the sizes typically found in the clinical setting for the tumor maximum sizes, which are around 100-120 cm$^3$. Hence, we selected $L = 80$ voxels per spatial length to make these sizes attainable. The time step was fixed at 4 hours. From typical cell sizes [27] we estimated the carrying capacity of a single voxel $N_{\max}$ to be $2 \times 10^5$ cells.

Sequencing studies of GBMs [14, 16] reveal that most of the mutations found in this disease cluster around three main pathways: RTK/PI3K/RAS, RB and p53. Each of these pathways has different key alterations and frequencies. We therefore considered a set of three possible alterations characterizing populations ($G = 3$), so that $\vec{g} = (g_1, g_2, g_3)$. This means that there are eight possible cell subtypes ($2^G$), depending on all possible combinations of altered pathways (Fig 2).

The choice of division, death, mutation and migration basal rates used a Bayesian criterion. To obtain an initial coarse estimate of the parameter ranges, the most straightforward way is to use imaging data from real GBMs. Basal reproduction and death rate (Table 1) were estimated

**Table 1. Parameter values for simulations of GBM.** Basal rates refer to cells with no mutations. Weights specify how each mutation affects the basal rate.

| Alteration | Reproduction | Migration | Mutation | Death |
|---|---|---|---|---|
| RTK | 0.32 | 0.65 | 0.18 | -0.15 |
| RB | 0.28 | 0.05 | 0.18 | -0.05 |
| p53 | 0.25 | 0.05 | 0.32 | -0.45 |
| Basal | 80-250 h | 0.0033-0.0125 mm$^2$/h | 80-240 h | 80-400 h |

from papers using exponential/gompertzian growth laws to fit GBM growth curves [28, 29]. Basal migration parameters were estimated from [29]. Since monitored tumors already carry an unknown mixture of alterations, these numbers should be taken only as rough estimates. Basal mutation rates were estimated from the known values per base and generation for each altered pathway [30–32]. To refine those ranges, thousands of simulations were run with input parameters randomly sampled from previous ranges. Simulations whose tumor lifespans were substantially longer than those typical of real GBMs were rejected. Simulations whose tumor lifespans were close to those of real GBMs were accepted. Basal rate ranges were thus constructed on input parameters from accepted simulations.

The next parameters to estimate are the mutation weights ($w_i$). Namely, the effect that alterations have on the rates at which cells perform basic cellular processes. Note that these weights are different from the mutation rates. We estimated mutation weights on a qualitative manner, based on the impact that alterations in considered pathways have over cell division, death, migration and mutation. For instance, alterations in RTK/PI3K/RAS pathway typically promote proliferation and migration. Cell cycle regulator p53 is mainly involved in genome repair, thus its alterations promote death evasion, as well as increased genetic instability, which leads to a higher mutation rate [33]. RB is another regulator that prevents excessive cell growth by inhibiting cell cycle progression, so alterations in its pathway will influence division and death as well. We chose a set of weights based on this biological knowledge, ensuring that clonal populations undergoing all mutations did not reach unrealistic proliferation rates, and we stuck to this set for all simulations performed throughout this study. The final choice of weights is shown in Table 1 as well as the chosen ranges for the values of the basal characteristic times. Notice that these times are associated with cellular processes; whole-tumor rates emerge as a result of combined cellular processes. Cellular traits were randomly sampled from the range of allowed basal rates for each simulation. This provided variability between individual simulations and allowed us to assess the robustness of the model's behavior.

Although the same set of mutation weights was used for all simulations, we assume that there exists a set of N probability distributions, one for each mutation weight, such that the alteration frequencies retrieved from in silico tumors resemble those of real tumors. To deepen our knowledge about mutation weight distribution and to improve their estimation, we used a simple version of Approximate Bayesian Computation (ABC) rejection algorithm [34] (see S2 Appendix). We run a number of simulations with a random sample of the mutational weights and accepted those values whose respective in silico tumor had a mutational landscape similar to that of real tumors [15, 16, 33, 35]. In this way, we can tune these weights to produce tumors with realistic mutational contents. Alteration frequencies of in silico tumors were retrieved at the time of diagnosis. We repeated this process with the posterior distribution until obtaining a robust estimate for the distributions. Note that our choice of mutation weights (Table 1), while being based on biological knowledge instead of ABC rejection algorithm, is pretty close to the most likely value from probability distributions of mutation weights obtained by this algorithm.

## Macroscopic tumor measures

We will use a set of measures to quantitatively compare tumor longitudinal dynamics with the solutions of the discrete simulation model. We list them here and give precise definitions.

**Heterogeneity.**    Diversity indexes are used in ecology to track genotype heterogeneity [36]. Two of the most frequently used are Shannon entropy and the Simpson index. The Shannon entropy quantifies the uncertainty in predicting the species of a selected individual in a

population.

$$H(t) = -\sum_{i=1}^{2^G} p_i(t) \log p_i(t), \tag{9}$$

where $p_i(t)$ is the proportion of individuals from species $i$ in the tumor at time $t$:

$$p_i(t) = \frac{\sum_{x=1}^{L_x} \sum_{y=1}^{L_y} \sum_{z=1}^{L_z} A_i(x, y, z, t)}{\sum_{x=1}^{L_x} \sum_{y=1}^{L_y} \sum_{z=1}^{L_z} \left( \sum_{i=1}^{2^G} A_i(x, y, z, t) \right)}, \tag{10}$$

where $A_i(x, y, z, t)$ is the number of active cells from species $i$ in voxel $(x, y, z)$ at time $t$. The Simpson index quantifies the probability of picking two individuals at random from the same subpopulation:

$$S(t) = \sum_{i=1}^{2^G} p_i^2(t). \tag{11}$$

A Shannon entropy equal to 0 means that all cells in the system belong to the same subpopulation, so there is no uncertainty in predicting cell type. A higher Shannon entropy means higher uncertainty and thus higher heterogeneity. A Simpson index of 1 indicates that all cells belong to the same type, while a value of 0 shows that there are no cells of equal type. In this study we were interested in heterogeneity dynamics of the whole tumor, so we considered Shannon entropy and the Simpson index integrated over all space, as functions of time. Using these indexes we can infer whether several cell populations with different mutational profiles coexist within the tumor, or a single cell population dominates over the others.

**Volumetric and morphological measures.** Let $\mathcal{V}$ be the set of voxels that have reached more than 20% of their carrying capacity, considering both living and necrotic cells. Let $\mathcal{V}_{CE}$ be the subset of $\mathcal{V}$ consisting of voxels in which active cells alone have reached more than 20% of the carrying capacity. Let $\mathcal{V}_I$ be the complementary subset $\mathcal{V}_I = \mathcal{V} \setminus \mathcal{V}_{CE}$. Let us define the number of elements in each set by $N_{CE} = |\mathcal{V}_{CE}|$, $N_I = |\mathcal{V}_I|$ and $N_T = |\mathcal{V}|$. Note that, because of this definition $\mathcal{V}_{CE} \cap \mathcal{V}_I = \emptyset$. Then, if individual voxel volume is $V_{vox}$, we define the contrast-enhancing ($V_{CE}$) and inner or necrotic ($V_I$) volumes as

$$V_{CE} = N_{CE} V_{vox}, \tag{12a}$$

$$V_I = N_I V_{vox}. \tag{12b}$$

Contrast-enhancing volume is associated with active tumor regions, while inner volume represents the necrotic core. Both quantities can be obtained from computer simulations of our mathematical model. The sum of both magnitudes represents the whole tumor volume, $V = V_{CE} + V_I$. The surface $\mathcal{S}$ enclosing $\mathcal{V}$, and its measure $S$, were obtained using the marching cubes method, seen in [24], to resemble the method used to extract this feature from MRIs.

We also defined the mean spherical radius (MSR), as the radius of a sphere having the same volume as the tumor, i.e.

$$\text{MSR} = \left( \frac{3V}{4\pi} \right)^{1/3}. \tag{13}$$

In addition to the volumetric measures we also employed several morphological descriptors that have been found to have prognostic value in different tumor types. They are active tumor spherical rim width ($\delta_s$) and surface regularity ($S_{reg}$). The first one is obtained from MRIs as an

averaged distance between the contrast-enhancing volume and the necrotic core [37]. It can be computed from the volumes through the formula

$$\delta_s = \left(\frac{3}{4\pi}\right)^{1/3}\left[(V_{CE} + V_I)^{1/3} - V_I^{1/3}\right]. \tag{14}$$

This biomarker has been found to have prognostic value for GBMs using both MRI [23, 37] and PET [38] images.

To quantify the surface regularity we used a dimensionless ratio defined as the relation between the total volume tumor $V$ and the volume of a sphere with the same surface $S$ [24]:

$$S_{reg} = 6\sqrt{\pi}\frac{V}{S^{3/2}}. \tag{15}$$

The closer this ratio is to 1, the more similar to a sphere a tumor will be (more regular). When $S_{reg}$ approaches zero the tumor will be highly irregular, resembling a fractal-like structure. This parameter receives different names in the literature and has been found to have prognostic value in lung cancer [39, 40], head and neck cancer [41, 42], esophageal cancer [43], breast cancer [44], lymphoma [45], and glioma [24, 38].

## Tumor growth law

The search for the mathematical equations that govern tumor growth has been a constant in the history of mathematical oncology [46–48]. Several attempts have been made to fit tumor growth laws to longitudinal volumetric data, including GBM [28, 49]. Here, we aim to reproduce this for the simulated tumors. We select exponential, gompertzian and radial growth, all of which have been analyzed in the previously cited studies. Motivated by the morphological analysis described above we also tried to fit to a power law, a variation of the von Bertalanffy equation and another usual candidate for tumor growth law. The equations to be fitted are therefore:

$$V(t) = V_0 \exp(\alpha t), \tag{16a}$$

$$V(t) = (4\pi/3)(r_0 + \alpha t)^3, \tag{16b}$$

$$V(t) = K \exp\left(\log(V_0/K)\exp(-\alpha t)\right), \tag{16c}$$

$$V(t) = [V_0^{(1-\beta)} - \alpha(\beta - 1)t]^{1/(1-\beta)}, \tag{16d}$$

where $V_0$ is initial volume and $r_0$ the corresponding mean initial radius, $\alpha$ denotes respective growth rates, $K$ is the carrying capacity in the Gompertz model and $\beta$ is the scaling exponent in the power law (it is positive and can be greater than 1, see [50]). We used Matlab function `lsqcurvefit` to obtain fitted parameters and root-mean-square error in order to compare goodness of fit. Initial value was fixed to that of the simulated tumor and the carrying capacity in the Gompertz model was assigned an upper bound of 1400 cm$^3$ (average cranial capacity). For the power law, we tried different scaling exponents $\beta$ and selected the one providing the best fit.

## Survival analysis

The measures explained above have prognostic value in real GBMs. In order to compare in silico and real tumors, survival time has to be modeled in some way. Given the current state of

the model, any definition of survival time will necessary be a simplification, since there are many factors that influence prognosis and fatality that cannot all be taken into account here. We follow other examples of survival analysis in mathematical models that use tumor size as an indicator of fatality [51, 52]. There is also clinical evidence that points to this association between prognosis and tumor size [53, 54].

We first define a diagnosis time. In order to do so, we have used data from 69 patients of The Cancer Imaging Archive, diagnosed with GBM (data available in S1 Table). In a previous work, these images were segmented and the total volumes computed [24]. We used these volumes to construct an empirical distribution of diagnostic sizes. As an independent validation of that empirical distribution, we also considered the patient data from [54]. There, it was found, in a cohort of 209 GBM patients, that the median preoperative tumor volume was 24.9 $cm^3$ (interquartile range 11.1-49.0 $cm^3$). This range agrees with the one used by us for constructing the distribution of volume sizes at GBM diagnosis. In contrast, this same procedure cannot be employed to retrieve a distribution for tumor volumes at patient's death. Usually the last available measure of tumor volume corresponds to the last follow-up, which differs from the moment of death to some extent. Most importantly, any empirical distribution of tumor volumes at the time of death would be biased by the treatment received by the patient. To avoid these limitations, we seized the estimate of 6 cm of diameter (113 $cm^3$) from [51, 52], which is based on studies of GBM sizes at exitus. In order to take into account the potential growth from the time of the last follow-up to decease, we select the range 100-120 $cm^3$. Having ranges of diagnostic and death sizes, we can assign a tumor lifespan to each simulation by sampling from the respective distribution.

We then performed a Kaplan-Meier analysis over sets of simulations to evaluate the model's ability to reproduce prognostic values. Kaplan-Meier is a non-parametric statistic that is used to estimate the survival function from a lifetime dataset. The survival function is a monotonically decreasing function that gives the probability of a patient/device's survival beyond a specified time. It is defined as follows:

$$S(t) = P(\tau > t); \ \tau \geq 0 \tag{17}$$

where $\tau$ is a random variable that represents the time that it takes for an event of interest to occur. Provided that we know the set of times $\{\tau_i\}$ at which the event of interest took place for each patient, the Kaplan-Meier statistic estimates the survival function as

$$\hat{S}(t) = \prod_{i:t_i \leq t} \left(1 - \frac{d_i}{n_i}\right) \tag{18}$$

where $t_i$ represents a time when at least one event took place, $d_i$ is the number of events that took place at time $t_i$ (death of a patient, in this context), and $n_i$ is the number of patients that are still alive at time $t_i$. As we have defined a survival time for each simulated tumor, we can calculate the Kaplan-Meier statistic to estimate the survival curve of a set of simulations.

A splitting threshold was used to separate simulations into two groups, according to the measures explained earlier in this section. The standard statistical test to compare Kaplan-Meier estimators from different groups is the log-rank test. Time separation between curves at median survival was also calculated. We performed a search over all possible splitting thresholds and assessed significance of each of them, provided that the larger group was not more than 3 times the size of the smaller group.

As survival time depends on random sampling of tumor volumes at diagnosis and at patient's death, we repeated this analysis 1000 times with different random seeds. Additionally, we performed this survival analysis twice, using two different distributions of tumor volumes

at death: a uniform distribution and a Gaussian one. In doing so we can check the effects that the selected distribution has in the survival times obtained. The whole virtual survival analysis is further detailed in S3 Appendix.

### Software

The model was coded both in Matlab (R2018a, The MathWorks, Inc., Natick, MA, USA) and Julia (version 1.1.1). The main workspace and simulation sections were coded in Julia, while data analysis and plotting were coded in Matlab. Simulations were performed on three different machines: a 12-core 32 GB RAM 2.7 GHz Mac Pro, a 6-core 16 GB RAM 3.7 GHz custom-built computer, and a 2-core 8 GB 2 GHz MacBook Pro. Computational cost per simulation was of the order of 3-5 minutes, depending on the machine used and the simulation parameters. More details about code and visualization can be found in the Github repository of the simulator (https://github.com/JuanJS117/MesoscopicModel).

## Results

Until now, we presented the mesoscopic model as a new methodology for tumor growth simulation at sizeable scale and in very short times. We now show in this section the functionality of the model. As a form of validation, we compare the clinically-sized simulated tumors to volumetric and morphological results of GBM. We also attempt to recover known evolutionary features of cancer such as heterogeneity, fixation and dominance of aggressive clones.

### Simulated tumors recapitulate known GBM timescales and resemble clinical imaging data

To simulate the basics dynamics displayed by the model, we ran a set of 100 simulations starting from one wild-type cell (i.e. without mutations) placed at the center of the spatial domain. Each simulation had a different set of basal rates, sampled randomly from the ranges specified in Table 1 as discussed previously. This allowed us to study variability in the tumor dynamics. Tumors evolved according to the rules explained in the previous section and the volumetric, morphological and metabolic macroscopic variables were tracked as discussed in the 'Materials and methods' section. Results are shown in Fig 3. The range of basal rates considered allows for the appearance of tumors with long inception (up to 10 months) as well as rapidly growing cancers (less than 5 months combining inception and growth), both in terms of MSR (Fig 3E) and volume (Fig 3F). These values are in agreement with clinical experience since treated GBM patients have a median overall survival time of about 15 months [12].

The filling of a single voxel follows a dynamics resembling logistic growth. An initial fast-growing phase is followed by a peak in the number of active cells, which begin to decline due to saturation-induced death and migration to surrounding voxels (Fig 3B). While the total number of cells (both active and necrotic) tends to the carrying capacity in the long term, active cells decline to zero steadily as they die due to damage to the microenvironment. This can be seen in Fig 3A, where newborn cells reach a peak in activity, early in growth, and then decline steadily. The logistic-like dynamics within individual voxels does not imply that the global growth of the tumor is also bounded. This is because the available physical space around populated voxels allows for sustained growth in tumor volume (Fig 3C). We tried to fit this sustained growth to different growth laws as explained in the 'Materials and methods' section (Fig 3G). The best fit for the median run according to the RMSE corresponded to a power law with scaling exponent of $\beta = 1.21$ (Fig 3H).

Since space is discretized in hexahedral units, and resolution is low for small tumors, volume and surface calculations are not precise until tumor cells have spread to a large group of

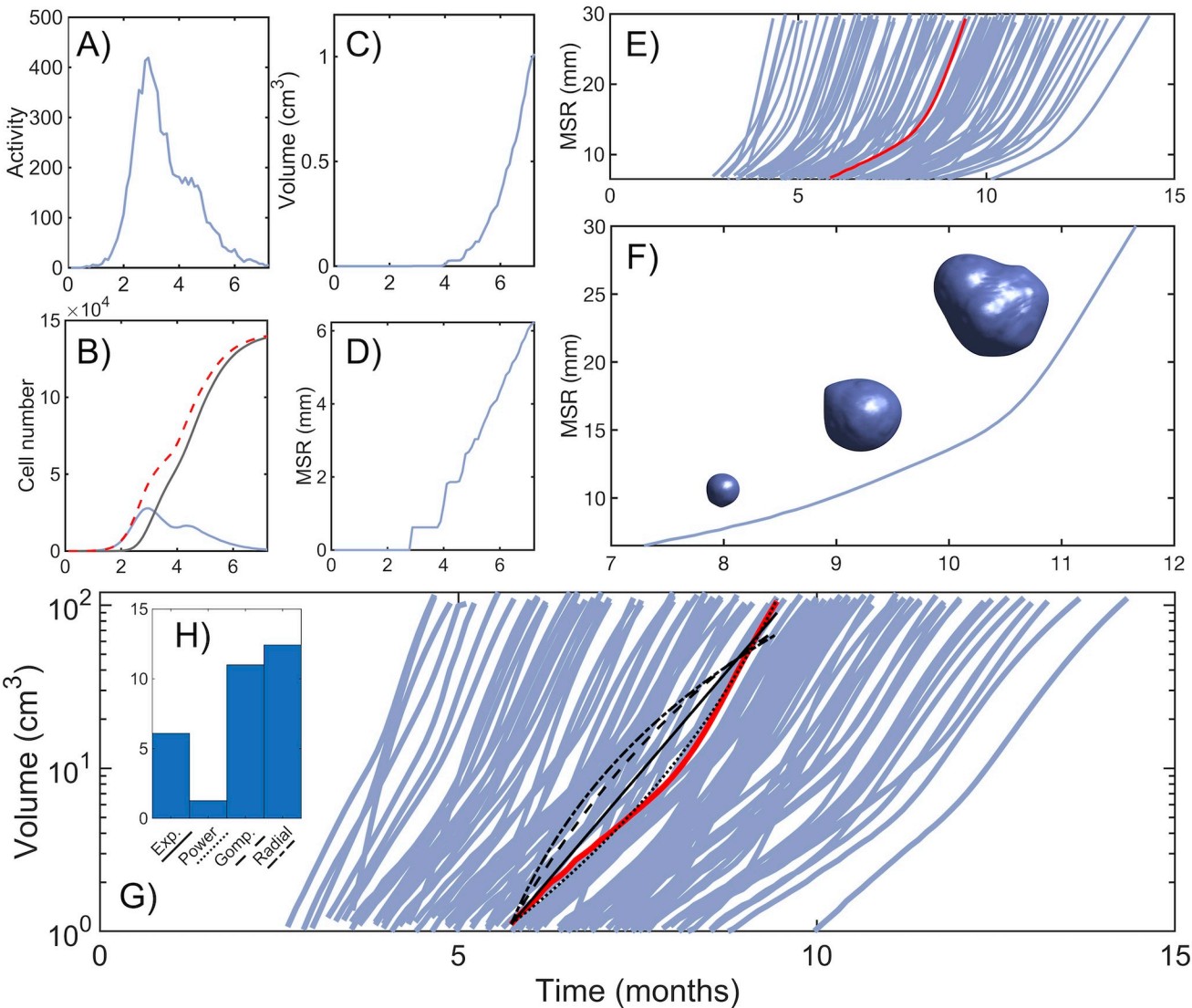

**Fig 3. Longitudinal tumor growth dynamics. A)** Time dynamics of the number of newborn cells at the central voxel (surrogate for tumor activity). **B)** Number of cells at the central voxel: total (dashed red), active (blue) and necrotic (grey). **C,D)** Tumor volume and MSR during the initial stages, starting from a single cell at the center of the lattice until tumor reaches 1 cm$^3$. **E)** Dynamics of MSR for 100 simulations. Median run is shown in red. Time span shown starts when tumor reaches 1 cm$^3$ in volume (equivalent to 6.2 mm of MSR) and ends after reaching 100 cm$^3$. **F)** Example of tumor dynamics of the MSR and rendered 3D tumor shape for three different times (8.5, 10, 11.5 months). Basal rate parameters for simulation in this figure are $\tau^{\text{rep}}$ = 216.5 h, $\tau^{\text{death}}$ = 112.7 h, $\tau^{\text{mut}}$ = 200.4 h, $\rho^{\text{mig}} = 0.0081 \frac{\text{mm}^2}{\text{h}}$. **G)** Dynamics of tumor volume for 100 simulations. Black lines represent different fits of the median run (solid red line): Exponential (solid), power law with $\beta$ = 1.2 (dotted), Gompertz (dashed) and radial (dashed-dotted). **H)** Root-mean-square error (RMSE) of each fit.

voxels. For example, if a tumor occupying three voxels expands to a fourth voxel, this would greatly distort its morphology, introducing fluctuations in morphological and geometrical measures. This is shown in Fig 3D, and especially affects those measures that approximate the tumor to a sphere (MSR, rim width, surface regularity). Because of this, tumor measures are only reliable when tumor volume exceeds 1 cm$^3$. Once this detectable size has been reached, MSR behaves linearly up to a point where there is acceleration in growth, due to the appearance and competition posed by more aggressive genotypes that increase the global growth rate. This is more clearly seen in Fig 3F, which shows the evolution of the MSR along with three-

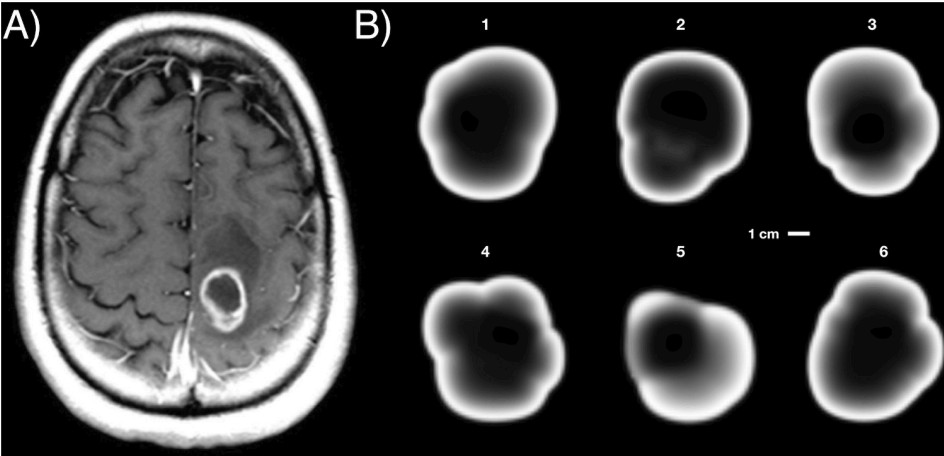

**Fig 4. Tumor slice and simulated profiles. A)** Two-dimensional slice of a T1-weighted post-contrast MRI scan of an actual GBM. **B)** Two-dimensional slices of different simulated tumors. Each simulated tumor corresponds to the final state (100 cm$^3$) of a different running of the model. Basal parameters for each simulation are: **1)** $\tau^{\text{rep}}$ = 242.6 h, $\tau^{\text{death}}$ = 213.27 h, $\tau^{\text{mut}}$ = 221.6 h, $\rho^{\text{mig}}$ = 0.0054 $\frac{\text{mm}^2}{\text{h}}$, **2)** $\tau^{\text{rep}}$ = 206.7 h, $\tau^{\text{death}}$ = 90.5 h, $\tau^{\text{mut}}$ = 219.5 h, $\rho^{\text{mig}}$ = 0.0044 $\frac{\text{mm}^2}{\text{h}}$, **3)** $\tau^{\text{rep}}$ = 184.2 h, $\tau^{\text{death}}$ = 325.4 h, $\tau^{\text{mut}}$ = 186.5 h, $\rho^{\text{mig}}$ = 0.0038 $\frac{\text{mm}^2}{\text{h}}$, **4)** $\tau^{\text{rep}}$ = 233.5 h, $\tau^{\text{death}}$ = 143.0 h, $\tau^{\text{mut}}$ = 175.9 h, $\rho^{\text{mig}}$ = 0.0052 $\frac{\text{mm}^2}{\text{h}}$, **5)** $\tau^{\text{rep}}$ = 201.2 h, $\tau^{\text{death}}$ = 295.5 h, $\tau^{\text{mut}}$ = 132.3 h, $\rho^{\text{mig}}$ = 0.0048 $\frac{\text{mm}^2}{\text{h}}$, **6)** $\tau^{\text{rep}}$ = 219.8 h, $\tau^{\text{death}}$ = 177.2 h, $\tau^{\text{mut}}$ = 87.33 h, $\rho^{\text{mig}}$ = 0.0042 $\frac{\text{mm}^2}{\text{h}}$. Other parameters are listed in Table 1.

dimensional reconstructions of the tumor for a typical run. The sharp increase in size occurs during the last two months of the disease, in agreement with the known lethal progression of these tumors [55].

Fig 4 shows two-dimensional slices of six simulated tumors, and one of a post-contrast pre-treatment T1-weighted MRI scan of a GBM patient. In this type of image, white areas correspond to regions where an intravenously injected gadolinium contrast is released into the tissue. The reason is that tumor blood vessels are less stable and lack functional pericytes. Thus, this marker is a surrogate of tumor cell density, leading to more abnormal vessels and suggesting that brighter areas would correspond to higher tumor cell loads. However, above a certain density, tumor cells damage the microenvironment and the secretion of prothrombotic factors leads to massive local cell death [26]. Inner dark regions represent necrotic tumor areas calculated as explained in 'Materials and methods'. The presence of a tumor-enhancing rim enclosing a highly irregular shape, which is a characteristic hallmark of GBM, is also captured in our simulations, and represents a region of highly proliferating cells [56].

## Evolutionary dynamics showed development of heterogeneity and dominance of the most aggressive clones

The simulator can be used to study the evolutionary dynamics within the tumor. An example is shown in Fig 5A, which shows the three-dimensional reconstruction of the tumor at three time points. In the first, the tumor mainly comprises the wild-type subpopulation, with small contributions of early-arising subtypes containing only one mutation. At the second time point we observe the emergence of more complex genotypes, with a significant contribution from the RTK mutated type, due to the effect of this mutation on proliferation. The end time point shows increased heterogeneity of the tumor with more altered genotypes taking over most of the tumor surface. This leads to the appearance of explosive peripheral areas (also resembling the lower parts of the tumor shown in Fig 4A). These features can be more clearly

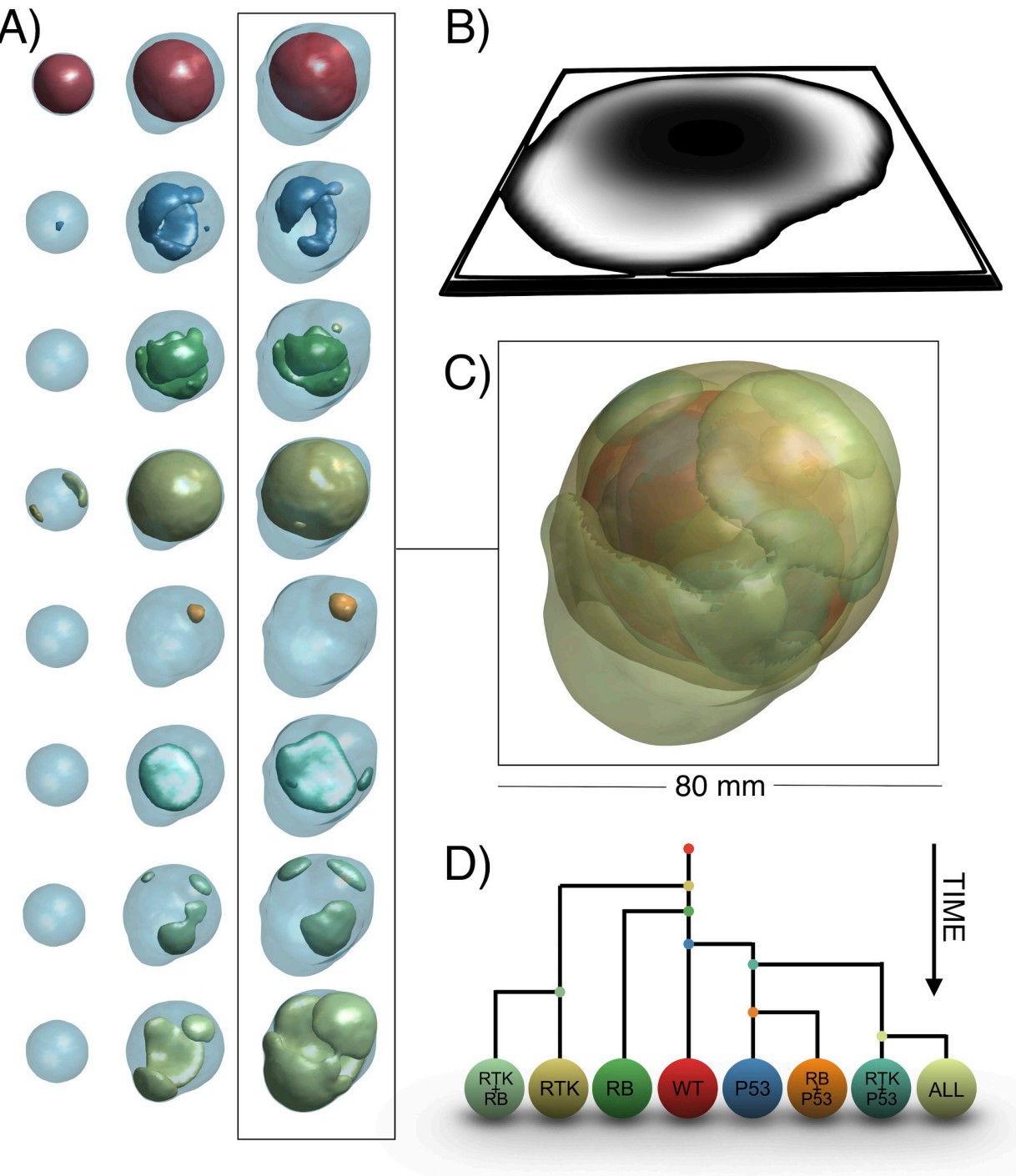

**Fig 5. Example of the dynamics of the tumor's clonal composition. A)** Evolution of the eight clonal populations included in the model (one per row). Total tumor volume is shown as a light blue background. Times correspond to 8.5, 10 and 11.5 months. Parameters for this simulation are $\tau^{\text{rep}} = 179.1$ h, $\tau^{\text{death}} = 292.5$ h, $\tau^{\text{mut}} = 222.7$ h, $\rho^{\text{mig}} = 0.0071 \frac{\text{mm}^2}{\text{h}}$. Other parameters are as in Table 1. **B)** Tumor central slice showing in white high tumor cell density. **C)** Three-dimensional subtype composition of the tumor. **D)** Reconstruction of the phylogeny of the tumor. Each bifurcation represents a mutation. Bifurcations occurring first in each branch represent mutations appearing earlier in time.

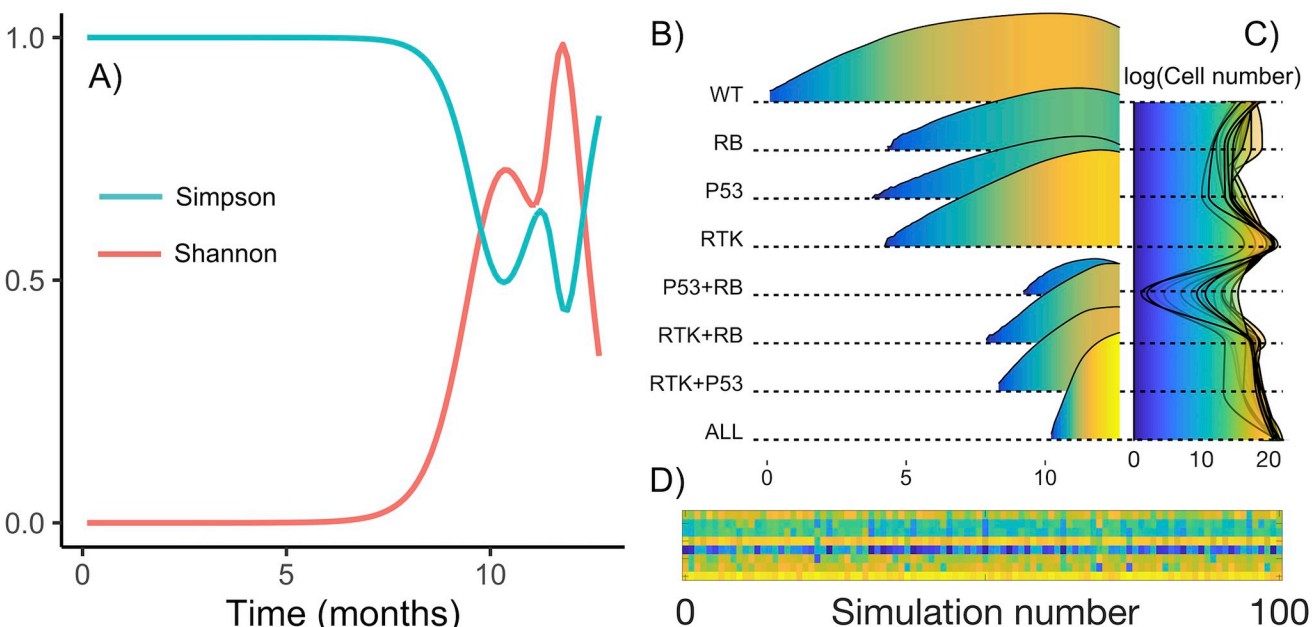

**Fig 6. Dynamical behavior of tumor heterogeneity. A)** Evolution of Shannon and Simpson's indexes for a typical run. Basal rates for this run are $\tau^{\text{rep}}$ = 242.6 h, $\tau^{\text{death}}$ = 213.3 h, $\tau^{\text{mut}}$ = 221.6 h, $\rho^{\text{mig}} = 0.0054 \frac{\text{mm}^2}{\text{h}}$. The other parameters are those of Table 1. **B)** Abundance of each subtype in logarithmic scale as a function of time. **C)** Superposition of final subtype abundance for all simulated tumors. **D)** Final abundance per subtype per simulation. Each row corresponds to one subtype, in the same order as above. Color indicates end-point abundance.

perceived in Fig 5B and 5C, a reconstruction of the whole tumor in its final stages. In this case tumor diameter is around 7 cm, again in the range of real GBMs [29, 37]. It is important to point out that the three-dimensional reconstructions are isosurfaces of the total volume occupied by a subpopulation. There is overlapping, since two subpopulations or more can coexist in a given region of the tumor, hence the superposition of colors seen in the reconstruction. Fig 5D shows a phylogenetic reconstruction of the tumor. This is typically done in genetic analysis of GBM [20, 21]. Here we show that it is possible in the model to track the time at which a given mutation appeared and reconstruct from which population it came. Note that phylogenetic tree reconstructions of clonal lineages for individual GBMs have been performed by combining bulk exome sequencing with single-cell RNA-seq data [57].

As new clonal subpopulations appear in the tumor, heterogeneity is expected to increase. However, due to cell competition and selection of the fittest, more aggressive subpopulations may end up occupying all the available space, confining less aggressive subpopulations to the core of the tumor and preventing them from proliferating, thus driving them to extinction. This fixation effect may result in a decrease in heterogeneity, as the fittest subpopulation becomes dominant. Fig 6A shows an example of the oscillatory behavior of the heterogeneity, as measured by the Shannon and Simpson's indexes. Changes in these indexes are associated with proliferation of several subtypes or the dominance of one, respectively. This is more clearly seen in Fig 6B, which depicts the abundance of each subtype on a logarithmic scale. In this example simulation there is a clear increase in heterogeneity between months seven and ten of the simulation, with the coexistence of the first four subtypes. The dominance of the RTK subtype then causes heterogeneity to decrease, only to rise again with the appearance of more complex genotypes. Expansions and extinctions seen in this figure are compatible with the reconstructions shown in Fig 5A. Depending on when and where new clonal subpopulations appear, each simulation will bring a different evolutionary dynamic. Often, the most

aggressive population carrying all alterations prevails, but a coexistence of two or more sub-populations may also take place. A combined view of the final state for all simulations is shown in Fig 6C. The heatmap below (Fig 6D) displays a clearer view of the possible endpoints of a simulation.

## Surrogates of tumor growth obtained from the model replicate the behavior of real GBMs

Much effort has been directed towards finding imaging-based prognostic biomarkers in GBM [17, 23, 24, 37, 58–62]. Our simulations allow the whole tumor natural history to be reconstructed, from its inception to the patient's death. We focus our attention here on variables that can be obtained from our simulations and that have been found to have a prognostic value: Rim width, and surface regularity (See 'Materials and methods').

One of the most typical analyses in terms of prognostic value involves extracting the values of these parameters from diagnostic images and correlating them with patient survival. In our case, the model allows for tracking the evolution of these metrics over the whole tumor life-span. Results for rim width and surface regularity for 100 simulations are shown in Fig 7. Time units have been normalized in order to compare simulations with different ranges of time evolution (see Fig 3). Curves represent the progression of the tumor from 1 cm$^3$ to 100 cm$^3$. Rim width typically increased with time as the tumor became more aggressive. This is an important difference with models having a simple clonal composition [56, 63], where the rim width was found to be constant. However, both approaches led to the same conclusion, namely, that rim width on diagnosis was associated with prognosis. Surface regularity was found to correlate with tumor clonal composition. Tumor slices corresponding to high and low values of each measure are also shown in Fig 7 to provide insight into their meaning.

As a test of the ability of our modeling methodology to replicate real tumor behavior we studied the association between the measures obtained from the simulations and overall survival, in silico. Overall survival was defined in terms of tumor burden (see 'Materials and methods'). Since our model did not include therapy, typical survival times are expected to be short. A summary of survival analysis results is shown in Fig 8, where it is clear that all measures showed statistically significant curve separation. Median differences were found to be small, due to lack of treatment, but highly significant. These results indicate that surface regularity and rim width had prognostic value in silico, as happens in real tumors [22–24, 50]. Poor prognosis was associated with larger rim widths and lower surface regularity.

## Mesoscopic simulation algorithm has good computational properties

One of the strong points of the mesoscopic simulation approach presented here is its low computational cost, even when working with high numbers of cells. Typical simulation times using the Julia code were a few minutes (2-6) in personal workstations and 80$^3$ grid points (Fig 9A). Although the velocity of the simulation increases with time and cell number (Fig 9B), timescales are still acceptable. A set of 100 simulations ran from the terminal sequentially takes around 5 hours. However, several terminals can be opened at the same time, which fastens simulations very much. Running 100 simulations simultaneously in 10 terminals (10 sequential simulations per terminal) takes 35 minutes, with a mean execution time of 31 min per 10 simulations.

Furthermore, the model has good parallelization properties. This is due to its discretization in spatial voxels that are updated independently and to the synchronous updating of the grid. It it also a common feature of lattice gas models, in which parallelization has already demonstrated increased computational performance [64]. Even though computational times are

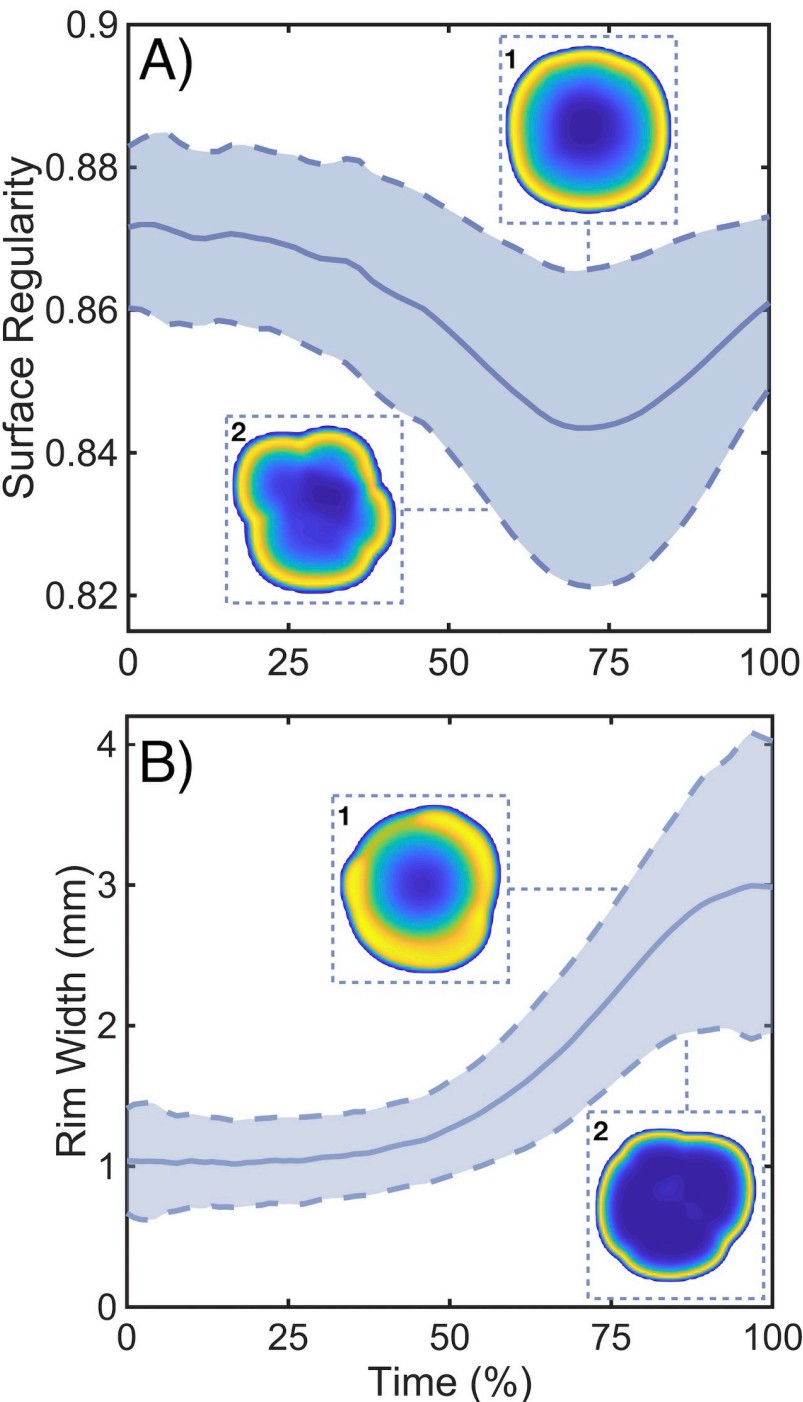

**Fig 7. Dynamics of prognostic measures obtained from the model. A,B)** Time evolution of spherical rim width and surface regularity for 100 simulations with parameters from Table 1. The solid line is the average value and the dashed line the standard deviation. 2D reconstructions correspond to characteristic upper and lower values of each variable. Basal parameters, measured in hours, for each subplot are: **A1)** $\tau^{\text{rep}}$ = 94.2 h, $\tau^{\text{death}}$ = 170.8 h, $\tau^{\text{mut}}$ = 99.4 h, $\rho^{\text{mig}} = 0.0034 \frac{\text{mm}^2}{\text{h}}$, **A2)** $\tau^{\text{rep}}$ = 184.9 h, $\tau^{\text{death}}$ = 230.1 h, $\tau^{\text{mut}}$ = 120.0 h, $\rho^{\text{mig}} = 0.0045 \frac{\text{mm}^2}{\text{h}}$, **B1)** $\tau^{\text{rep}}$ = 104.8 h, $\tau^{\text{death}}$ = 323.5 h, $\tau^{\text{mut}}$ = 222.7 h, $\rho^{\text{mig}} = 0.00231 \frac{\text{mm}^2}{\text{h}}$, **B2)** $\tau^{\text{rep}}$ = 158.7 h, $\tau^{\text{death}}$ = 54.2 h, $\tau^{\text{mut}}$ = 197.7 h, $\rho^{\text{mig}} = 0.005 \frac{\text{mm}^2}{\text{h}}$.

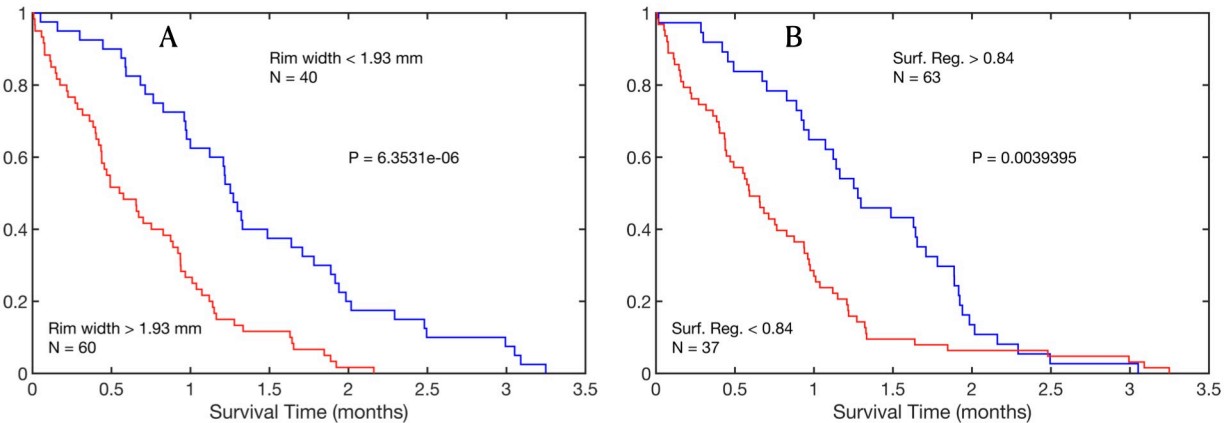

**Fig 8. Prognostic measures obtained from the mathematical model recapitulate the behavior of those obtained from MRI and PET images of GBMs. A).** Kaplan-Meier curves for the population splitting using the spherical rim width taking a threshold equal to 1.93 mm. Median survival difference between groups was 0.62 months. **B).** Kaplan-Meier curves for the population splitting using the surface regularity taking a threshold equal to 0.84. Median survival difference between groups was 0.58 months.

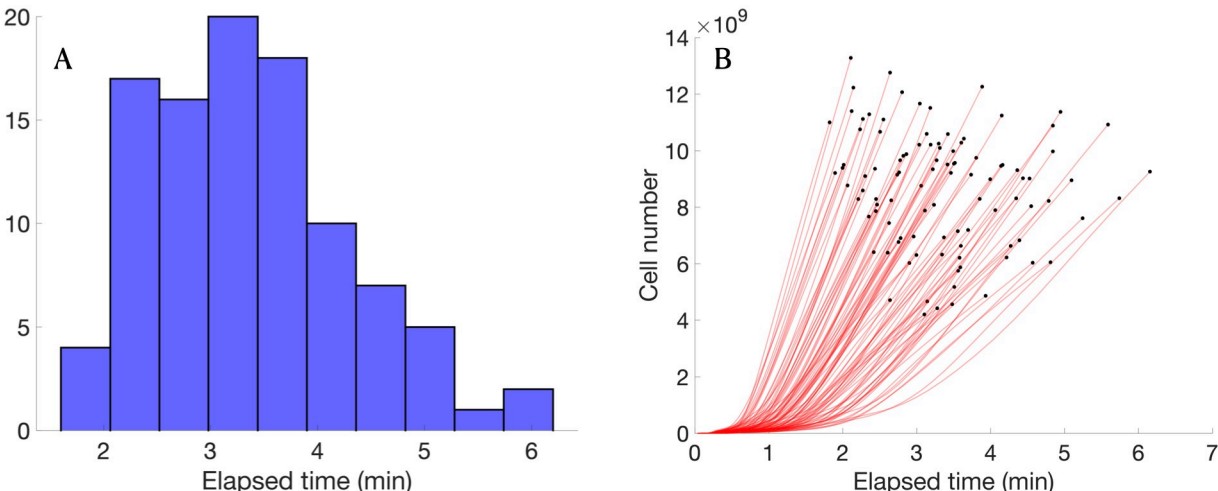

**Fig 9. Elapsed time of typical model simulations. A).** Elapsed simulation time for a set of 100 simulations ran independently. **B).** Comparison between elapsed simulation time and tumor cell number at different time steps for a cohort of 100 simulations. Cell number does not include dead cells.

acceptable at this stage of modeling, there are several actions, such as increasing the lattice size, adding biological processes or performing many runs to explore parameter regions, that may lead to substantially longer computation times. Parallelized versions of the code will be therefore developed alongside these additions.

## Discussion

In this paper we have presented a stochastic mesoscale simulator aimed at mimicking the natural history of a tumor from its inception to clinically observable sizes. Many different discrete simulation approaches are available to accomplish that task and shed light on different processes in oncology [1–6]. In our case the focus was on finding a balance between

computational complexity and biologically meaningful assumptions, allowing for the study of a number of macroscopic features over the whole lifespan of the malignancy.

The tumor was described at the mesoscopic scale as a composition of different clonal populations at the voxel level, each of them having clone-specific characteristics that determine their behavior. Cells grew by proliferation, migrated and diversified as a result of mutational events that altered specific cellular processes. Death accounted for cell turnover and necrotic core formation. These biological rules were implemented as probabilistic events, incorporating both internal and external influences. From the point of view of simulation, these rules were set up on a discrete three-dimensional space, following the perspective of multi-compartmental cell automaton models and matching the spatial resolution to current high-resolution medical imaging standards. The three-dimensional lattice was updated synchronously, taking the initial cell to a fully grown, spatially heterogeneous tumor. With efficiency in mind, the goal of this setup was to minimize computational time. This makes it possible to use our modeling framework to rapidly study different tumor dynamics scenarios and to test novel hypotheses. The algorithm allows for further improvements in speed by adding parallelization. Special emphasis was placed on the generation of a context of heterogeneity, competition, fixation and natural selection, a contribution framed in the mathematical modeling of evolutionary processes [65–67], rather than accurately parameterizing the model.

Other discrete modeling paradigms like those based on the Gillespie algorithm [68] provide correct solution trajectories for stochastic processes but become inefficient and computationally intensive when the number of events is high, due to the tau-leaping updating method. Continuous models have been also employed in the study of evolutionary dynamics. For example, partial differential equations can be obtained by means of mean field approximations of the individual-level events. This analysis, however, can potentially neglect spatial correlations between the locations of individuals, specially when different species or subpopulations are taken into account [69]. In circumstances where the mean-field approximation is unsuitable, one needs to include information on the spatial distributions of individuals (or subpopulations of those individuals), which is not a simple task from the continuous point of view. On the contrary, this characterization can miss features that are only brought to attention by the combination of discrete, spatial and stochastic dynamics [70].

Dynamical behavior of the tumor was first analyzed in terms of volume and radial growth. Despite the logistic nature of each voxel's dynamics, the whole tumor showed sustained growth, first linear and then accelerating as a result of the diversification and interplay of the populations, in a process that selects for more aggressive clones. Curve fitting resulted in power law being the most accurate description, over other unbounded laws like exponential or linear radial, pointing towards a relationship between metabolic activity, evolutionary dynamics and aggressiveness [50]. The dynamics of simulated tumors changed from run to run as a result of the stochastic nature of the model, which allowed the influence of one-off events and parameter variability to be studied. [70]. The model provided a framework to analyze situations of clonal evolution of populations that are otherwise inferred from measurements of mutational spectrum and proportions [19–21, 71]. Furthermore, the capacity to extend this longitudinal simulation up to clinical sizes permits comparison of the dynamics with medical imaging, which can in turn be used in model calibration and quantitative description [72]. This opens the door to the analysis of other features such as the expansion and size of the necrotic core and its dependence on cell death and turnover.

Population diversification is another fundamental feature of the simulation and is analyzed here from an evolutionary ecology point of view, with metrics like the Shannon and Simpson indices [36]. This has already been done in heterogeneity analysis of tumors [73, 74]. The observed oscillations in such indexes reveal the process of clonal evolution and population

fixation, in which aggressive phenotypes progressively displace the previous clones. Heterogeneity decreases when this happens and is maintained when competition is active in different regions of the tumor. This is interesting *per se* since heterogeneity is also spatially distributed, meaning that there may be spatial areas of the tumor with more diversity than others. A whole-tumor measure of diversity misses these characteristics. Three-dimensional reconstructions and exploration of the distribution of clonal populations enables the exploration of such scenarios. Again, the possibility of understanding this process longitudinally can be a source of hypothesis testing, particularly when combined with RNA-sequencing techniques to reconstruct phylogenetic trees of clonal lineages for individual GBMs. Moreover, our framework can readily incorporate the action of chemotherapeutic agents and capture the emergence of the different processes contributing to drug resistance [75], and could also be useful to find better chemotherapy schedules [76].

Using the model, we studied two macroscopic quantities that have been proven to show prognostic value: surface regularity and tumor rim width [22–24]. The model simulations were able to reproduce the behavior of these significant metrics having clinical value. Time evolution curves of these two variables showed a progressively increasing rim width up to a saturation value and a decrease in sphericity. Both can be explained as a result of the appearance of more malignant clones that take over at the boundary, giving way to a larger infiltrative area which manifests geometrically as a lobule protruding from the main tumor mass. This association between degree of malignancy in terms of tumor size and morphology was confirmed by the survival analysis of both variables.

As with any simplified dynamical model in the biological sciences, in its simplicity lie both its virtues and its drawbacks. The downsides are the low spatial resolution at early stages due to the coarse discretization; the absence of a microenvironment, its constituents and their influence on cellular processes and the lack of a clear distinction of genotype and phenotype in order to study the connection between them. The lack of these elements impedes performing a survival analysis that takes into account all factors involved in prognosis, being limited here to the correlation between decease and tumor burden. Also, the number of clones is specified beforehand, in contrast to other evolutionary approaches in which the main elements emerge from a more simplistic consideration of features (see [9] for a comprehensive review).

These drawbacks give us the future lines of work with the mesoscale simulator. A first line of action is the addition of external components such as stromal cells, nutrients and vasculature. This is of utmost importance since many therapies are critically dependent on the tumor vasculature status. Stromal and other cell types can be added by including an additional variable in each voxel, modifying the probability computation according to their influence on a given process. Nutrients are normally modeled by means of diffusion and consumption equations, which can be integrated in discrete models by discretization [77]. Finally, vasculature can also be considered by embedding a network into the spatial grid [78]. A second line of action is to use the model for predicting patient's outcome. In this regard, we are in the process of building a database of GBM patients with genetic, geometrical, morphological and clinical information about each patient. This kind of information is also available in online databases such as The Cancer Genome Atlas program. These data will allow us to more precisely portrait the genetic landscape of individual patients, adapt the parameter estimation presented here and be able to predict relationships between phenotypic characteristics and geometrical features. Finally, the last line of action is the implementation of treatment in the model. Treatment can be included as a new event, producing cell death and inducing a deceleration on cell division. The dynamics of this treatment-induced cell death depend on the type of therapy considered. For example, radiotherapy would require considering a specific dose and a tumor radiosensitivity, while chemotherapy would require addressing dose effects, half maximal

effective concentration (EC50), and genetic/phenotypic cell traits related to drug resistance. A more elaborated but not unaffordable approach would consist of including immune cells as a new population, and all the interactions involved between them and tumor clonal populations, so that immunotherapy could also be introduced in the model. Ongoing work is being done in implementing both radiotherapy and chemotherapy treatments for glioblastoma, with temozolomide as the drug of choice.

The ultimate goal of the mesoscopic model is to provide a readily usable framework not only for hypothesis testing, but also for outcome prediction. Fitting tumors of individual patients is admittedly a difficult task for any model, as it would require obtaining patient-specific parameter values from patient data, and available data from a single patient is usually scarce. Rather than individual patient's outcome prediction, our aim is to perform broad computational studies, such as best outcome search from alternative treatment protocols, by fitting model parameters over large sets of patients. This will help improving current treatment schedules, which in turn may result in a better quality of life for cancer patients. We hope this new methodology will be found to be a useful addition to the plethora of discrete simulation approaches intended to benefit cancer patients through the tools that computational approaches can provide.

## Supporting information

**S1 Appendix. Migration process.**
(PDF)

**S2 Appendix. Bayesian optimization of mutation weights.**
(PDF)

**S3 Appendix. Robustness of virtual survival analysis.**
(PDF)

**S1 Table. Data from TCGA patients.**
(XLSX)

## Author Contributions

**Conceptualization:** Juan Jiménez-Sánchez, Álvaro Martínez-Rubio, Youness Azimzade, Juan Belmonte-Beitia, Gabriel F. Calvo, Víctor M. Pérez-García.

**Formal analysis:** Juan Jiménez-Sánchez, Álvaro Martínez-Rubio.

**Funding acquisition:** Víctor M. Pérez-García.

**Investigation:** Juan Jiménez-Sánchez, Álvaro Martínez-Rubio, Gabriel F. Calvo, Víctor M. Pérez-García.

**Methodology:** Juan Jiménez-Sánchez, Álvaro Martínez-Rubio, Anton Popov, Julián Pérez-Beteta, David Molina-García, Juan Belmonte-Beitia, Gabriel F. Calvo, Víctor M. Pérez-García.

**Project administration:** Víctor M. Pérez-García.

**Resources:** Julián Pérez-Beteta, David Molina-García, Víctor M. Pérez-García.

**Software:** Juan Jiménez-Sánchez, Álvaro Martínez-Rubio, Anton Popov, Julián Pérez-Beteta, David Molina-García.

**Supervision:** Juan Belmonte-Beitia, Gabriel F. Calvo, Víctor M. Pérez-García.

**Validation:** Víctor M. Pérez-García.

**Visualization:** Juan Jiménez-Sánchez, Álvaro Martínez-Rubio.

**Writing – original draft:** Juan Jiménez-Sánchez, Álvaro Martínez-Rubio.

**Writing – review & editing:** Juan Jiménez-Sánchez, Álvaro Martínez-Rubio, Anton Popov, Julián Pérez-Beteta, Youness Azimzade, David Molina-García, Juan Belmonte-Beitia, Gabriel F. Calvo, Víctor M. Pérez-García.

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
