## [Decision Letter · Decision Letter 0]

2 Oct 2020

Dear Mr. Jiménez-Sánchez,

Thank you very much for submitting your manuscript "A mesoscopic simulator to uncover heterogeneity and evolutionary dynamics in tumors" for consideration at PLOS Computational Biology.

As with all papers reviewed by the journal, your manuscript was reviewed by members of the editorial board and by several independent reviewers. In light of the reviews (below this email), we would like to invite the resubmission of a significantly-revised version that takes into account the reviewers' comments.

We cannot make any decision about publication until we have seen the revised manuscript and your response to the reviewers' comments. Your revised manuscript is also likely to be sent to reviewers for further evaluation.

Sincerely,

Philip K Maini

Associate Editor

PLOS Computational Biology

Douglas Lauffenburger

Deputy Editor

PLOS Computational Biology

Reviewer's Responses to Questions

**Comments to the Authors:**

Reviewer #1: PCOMPBIOL-D-20-01441 A mesoscopic simulator to uncover heterogeneity and evolutionary dynamics in tumors

The authors present a computational framework to model cancer progression. The authors present the mathematical rationale for the various components of their simulator, and demonstrate several of the concepts and motivations of their work through an application to modeling the primary brain tumor glioblastoma. The manuscript is very well crafted, with very clear writing and visually striking figures. The authors cite much of the relevant literature, and discuss the limitations and potential future applications of their simulator. However, the authors include no data to support or validate their model. This omission leaves this reviewer confused as to whether this is a methods or results paper. The intent of the work is important in critically evaluating the suitability for publication.

As a methods paper, there are major concerns regarding the functionality of the codes, unit testing, comparison to other platforms, and computational benchmarking.

As a results paper, there are major concerns regarding validating and/or justifying many of the significant assumptions of the modeling platform, and reproducing the results.

Major concerns

>>The first and most important concern is to identify what is the intent of this manuscript. Is this a methods paper for a computational platform? Or a results paper?

As a methods paper,

>>the codes were provided as a URL to a zip file. This reviewer was unable to run the code with default installation settings in Julia. The code should be shared on a public repository such as github with more thorough documentation.

>>The authors use very specific probability distributions extensively throughout their approach, but provide almost no justification or rationale for their choices. Moreover, the authors do not demonstrate the sensitivity of their method/approach/results to different choices of distributions, ex. Pg 4 lines 108-109.

>>pg 5 line 173, What are “updating artifacts” and what is the “auxiliary structure” that mitigates them? This is an important detail to clarify, particularly for a methods paper.

>>Survival analysis is dramatically oversimplified with fatal tumor burden. It is misleading to present results that imply patient survival is related to this simulation measure. This should be addressed in the Discussion and clearly stated in the Methods. This reviewer recommends--but does not insist upon--removing this analysis all together. Also a justification and reference are required for the 100cm^3

>>For code parallelization, the authors test only 2 and 4 core configurations and show in Figure 9 a non-linear relationship. The authors should test on more cores and show results. Also, how do the authors establish standard deviation for the reference 1 core simulation and why is the ST the same for all cores? These kinds of details are important for a methods paper and are effectively irrelevant for a results paper.

As a results paper,

>>How are the authors handling the effects of mutations on cell dynamics. ? The authors appear to admit that this is not measurable on pg 7 lines 202-204. If this is the case, how can the authors claim anything as a result which rests on this assumption?

>>pg 7 line 214. This reviewer commends the authors for the intent to “make the study reproducible.” Unfortunately however, even if the simulation codes could be run (see earlier comment), it is not clear from the very brief code documentation how the results of this study could be reproduced.

>>pg 10 lines 300, 333 the authors refer to MATLAB code, but do not provide these codes. This is important for both results and methods views of this manuscript.

>>How do the authors propose to include normal tissue or stromal cells in the computational model? This is particularly important for GBM. The authors should discuss extensions to this frameworks to account for such details in future work. Also potentially aspects of the immune system and treatment.

>>The simulation results do not offer much insight or new findings in the “evolutionary dynamics in tumors” as the title suggests. Heterogeneity, fixation, and dominance of most aggressive clones are well known results in cancer evolutionary theory.

>>pg 19 line 520 What simulation results do the authors have to claim “...competition led to natural selection...”

Minor concerns and typographical errors

>>pg 3 ln 55; tumor treating fields have shown a significant improvement in GBM survival since the Stupp protocol of 2005 and should be noted

>>pg. 8 line 234 should be “A higher *Simpson index…”

>>where does the value of 0.62 in equation 13 come from?

>>in silico is sometimes hyphenated and sometimes not.

>>Figure 8. “Rim width” is reported to 4 significant figures of mm, which is not measurable on MRI. Report all measurements to realistic, measurable resolution.

>>Figure 9 x-axis should be in integer intervals.

>>pg 19 line 532 “variability to *be studied.”

Reviewer #2: The review is uploaded as an attachement.

Reviewer #3: Summary and General Feedback

In this manuscript, the authors formulate a mescoscale cellular automaton mathematical model describing the temporal and spatial evolution of tumour growth and its associated genetic composition. This model incorporates cell proliferation, death, migration, and mutations based on fixed-time probabilitistic events, allowing several events to occur in a single time step. Emphasis is given to fast computation time and ability for code parallelisation, which is a crucial component of large-scale simulations such as those performed in this manuscript. Good agreement is seen between model simulations and experimental evidence, both in terms of tumour shapes and associated growth dynamics. The model simulations predict heterogeneity in terms of tumour gene expressions, though in general, one particular gene expression tends to dominate in a local region of the tumour. Finally, a discussion concerning prognostic measures and survival analysis demonstrates a clear division between hypoactive and hyperactive tumour growth, based on typical parameter values obtained in the literature.

This paper is well written and very clearly explained. I’m impressed with the amount of novel results packed into this paper and therefore have minimal comments to suggest. One item that I do note is that some parts of the manuscript (e.g. the Discussion), while clear and informative, are long. While not essential, I would recommend some trimming of some sentences to reduce the over page count.

Minor Comments

Line 107: you have the inequality constraint that the sum of w_i is less than or equal to 1. What happens if this sum is equal to 1? Based on equation (2), and provided that the g vector is the ones vector, this means that tau_rep=0, which I don’t understand how that can be used in P_rep calculations in equation (1). A similar question arises in equation (4) when sum of w^mig =1.

Line 109: replace ‘Montecarlo’ with ‘Monte Carlo’

Line 253: Is V=V_CE + V_I?

Line 306: While Kaplan-Meier curves are mentioned in the Methods section and used in the Results (e.g. Figure 8), I don’t see a mathematical definition of these survival curves listed in the Methods. Similar to how you’ve stated the Shannon and Simpson indices, I think that included a brief mathematical statement of what you’re computing in Figure 8 would add some clarity.

**Have all data underlying the figures and results presented in the manuscript been provided?**

Reviewer #1: Yes

Reviewer #2: Yes

Reviewer #3: None

PLOS authors have the option to publish the peer review history of their article (what does this mean?). If published, this will include your full peer review and any attached files.

Reviewer #1: No

Reviewer #2: No

Reviewer #3: **Yes: **Nabil T Fadai
---

## [Decision Letter · Decision Letter 1]

16 Jan 2021

Dear Mr. Jiménez-Sánchez,

We are pleased to inform you that your manuscript 'A mesoscopic simulator to uncover heterogeneity and evolutionary dynamics in tumors' has been provisionally accepted for publication in PLOS Computational Biology.

Best regards,

Philip K Maini

Associate Editor

PLOS Computational Biology

Douglas Lauffenburger

Deputy Editor

PLOS Computational Biology

Reviewer's Responses to Questions

**Comments to the Authors:**

Reviewer #1: All concerns have been thoroughly addressed.

Reviewer #2: The authors have considered all my comments and have answered all my questions. They improved the article by adding explanations and details in the text.

The article is now suitable for publication in PLOS Computational Biology.

Reviewer #3: I thank the authors for carefully addressing my comments in my previous review; I believe that this revised manuscript is suitable for publication.

**Have all data underlying the figures and results presented in the manuscript been provided?**

Reviewer #1: Yes

Reviewer #2: Yes

Reviewer #3: None

PLOS authors have the option to publish the peer review history of their article (what does this mean?). If published, this will include your full peer review and any attached files.

Reviewer #1: No

Reviewer #2: No

Reviewer #3: **Yes: **Nabil T Fadai

---

## [Editor Report · Acceptance letter]

4 Feb 2021

PCOMPBIOL-D-20-01441R1 

A mesoscopic simulator to uncover heterogeneity and evolutionary dynamics in tumors

Dear Dr Jiménez-Sánchez,

I am pleased to inform you that your manuscript has been formally accepted for publication in PLOS Computational Biology. Your manuscript is now with our production department and you will be notified of the publication date in due course.

With kind regards,

Alice Ellingham
